# Enhanced Integrity of White Matter Microstructure in Mind–Body Practitioners: A Whole-Brain Diffusion Tensor Imaging Study

**DOI:** 10.3390/brainsci13040691

**Published:** 2023-04-20

**Authors:** Yingrong Xie, Kelong Cai, Jingang Dai, Gaoxia Wei

**Affiliations:** 1CAS Key Laboratory of Behavioral Science, Institute of Psychology, Beijing 100101, China; 2Department of Psychology, University of Chinese Academy of Sciences, Beijing 100101, China; 3College of Physical Education, Yangzhou University, Yangzhou 225127, China; 4Experimental Research Center, China Academy of Chinese Medical Sciences, National Chinese Medicine Experts Inheritance Office of Song Jun, Beijing 100700, China

**Keywords:** diffusion tensor imaging, factional anisotropy, Tai Chi Chuan, white matter

## Abstract

Tai Chi Chuan (TCC) is an increasingly popular multimodal mind–body practice with potential cognitive benefits, yet the neurobiological mechanisms underlying these effects, particularly in relation to brain white matter (WM) microstructure, remain largely unknown. In this study, we used diffusion tensor imaging (DTI) and the attention network test (ANT) to compare 22 TCC practitioners and 18 healthy controls. We found extensive differences in fractional anisotropy (FA), mean diffusivity (MD), axial diffusivity (AD), and radial diffusivity (RD) between the two groups. Specifically, TCC practitioners had significantly different diffusion metrics in the corticospinal tract (CST), fornix (FX)/stria terminalis (ST), and cerebral peduncle (CP). We also observed a significant correlation between increased FA values in the right CP and ANT performance in TCC practitioners. Our findings suggest that optimized regional WM microstructure may contribute to the complex information processing associated with TCC practice, providing insights for preventing cognitive decline and treating neurological disorders with cognitive impairment in clinical rehabilitation.

## 1. Introduction

Tai Chi Chuan (TCC), which originated in ancient China, is a multimodal mind–body exercise that integrates physical, cognitive, social, and meditative components during practice [1]. Numerous benefits of TCC practice include improving muscle strength [2], flexibility [3], and balance and motor control [4,5], as well as reducing the risk of falling [6] and alleviating pain [7]. Moreover, increasing evidence indicates that TCC could act as a buffer against cognitive decline through a diverse range of neural pathways. For instance, Yu et al. [8] compared TCC with conventional exercise and found that the two exercise modalities manifested different effects on global cognitive function, affected cognitive domains, and had different improvement rates. The underlying mechanism was that intricate motor training in conjunction with meditation in TCC might lead to earlier and more significant improvements in cognitive flexibility. A systematic review and meta-analysis revealed the potential of TCC to enhance cognitive function, particularly executive function, among healthy older adults [1]. Our previous study demonstrated that TCC practitioners showed shorter reaction times (RT) in a flanker test than the control group [9]. The benefits of TCC on cognition have been supported by clinical trials. For instance, as a type of complementary and alternative medicine, TCC could significantly induce improvements in cognitive function in patients with Parkinson’s disease [10], Alzheimer’s disease [11], multiple sclerosis [12], cardiovascular diseases [13], and mood disorders [14].

Although an abundance of studies have focused on the cognitive gains of TCC, the potential neural mechanism has not been fully explored. Being an integrative practice of mental and physical exercise, TCC may induce reliable and optimized changes in brain structure associated with improved cognition. Supporting this speculation, our previous neuroimaging study [15] demonstrated that TCC training induced decreases in functional homogeneity (improved functional specialization) in the left anterior cingulate cortex and increases in functional homogeneity (improved functional integration) in the right post-central gyrus. Both of these changes predicted performance gains on attention network behavior tests, providing evidence for the functional plasticity of the brain’s intrinsic architecture toward optimizing local functional organization.

White matter (WM) plays a key role in cognitive processes since it connects different grey matter regions throughout the human brain [16]. Increasing evidence has shown that impaired WM is associated with a decline in cognitive performance, including executive functions, working memory, and information processing speed [17,18,19]. For instance, age-related reductions in WM integrity can result in a disconnection state that underlies some age-related performance declines on tasks [20]. On the other side, heavily myelinated fibers conduct neural signals with high velocity, resulting in better cognitive functions [21]. Furthermore, WM abnormalities are often observed in some neurological diseases [22]. Diffusion tensor imaging (DTI) is an in vivo neuroimaging technique that provides a valuable tool for characterizing WM microstructural changes. It is an MRI modality that measures water diffusion in multiple directions to probe the structural and functional properties of biological tissues at the microstructural level. The most commonly used DTI quantitative indices are fractional anisotropy (FA) and mean diffusivity (MD), which can assess alterations in water molecule diffusivity and tissue structural integrity. Increased FA is often interpreted as reflecting a higher degree of WM integrity [23], while lowered FA has been observed in various conditions associated with fiber integrity loss, such as Alzheimer’s disease [24]. WM integrity can be further characterized by measuring the rate of diffusion along the primary (axial diffusivity, AD) and secondary (radial diffusivity, RD) axes of diffusion ellipsoids. AD is generally related to axonal integrity and thus decreases in cases of axonal damage [25], whereas RD appears to be modulated by myelin in WM. For example, increases in RD have been linked to myelin degeneration or loss [26]. RD can also be influenced by the diameter and density of axons [27]. A growing body of evidence suggests that TCC is associated with WM alternations. For instance, long-term TCC practitioners have been reported to have higher FA in the splenium of the corpus callosum [28]. However, it remains largely unknown what neural mechanism underlies the improved cognition induced by TCC from the perspective of WM integration.

The aim of this study was to investigate potential changes in the intrinsic architecture of the human brain associated with TCC practice, as well as any relevant cognitive gains. Accordingly, we compared a group of highly experienced TCC practitioners to a healthy control group to determine whether any differences in WM structure exist. We further hypothesized that changes in WM microstructure would be related to improvements in executive function.

## 2. Materials and Methods

### 2.1. Participants

A total of forty participants were involved in this study. Twenty-two TCC practitioners with TCC experience (age: 52.4 ± 6.8 years; 7 males, 15 females) were from the local TCC fitness center, while eighteen controls (age: 54.8 ± 6.8 years; 8 males, 10 females) were from the local community, with the criteria including having no history of regular physical exercise, yoga, or meditation within the past ten years. On average, the TCC participants had 14.6 ± 8.6 years of TCC experience; considering the different practice intensities, the mean hours of TCC experience is 9157 ± 7617 h. As to the education level, the TCC group has a mean value of 12.2 ± 2.9 years, while the control group is 11.8 ± 2.9 years.

All participants were screened and completed the screening forms to make sure they met the criteria for MRI scanning. The participants with histories of hearing or vision damage, physical injury, seizures, metal implants, or head trauma with loss of consciousness were excluded. All the participants completed a basic demographic questionnaire. This study was conducted in accordance with the Declaration of Helsinki and its later amendments and was approved by the Institutional Review Board of the Institute of Psychology, Chinese Academy of Sciences (CAS). Written informed consents were obtained from all participants.

### 2.2. MRI Acquisition

Diffusion tensor imaging (DTI) measurements were performed on a 3T Trio Tim scanner (Siemens, Erlangen, Germany) with a 12-channel head matrix coil. All diffuse weighted images were obtained using a single shot echo-planar imaging sequence with the following scan parameters: diffuse direction = 64, TR = 6600 ms, TE = 104 ms, flip angle = 90°, slices = 45, slice thickness = 3.0 mm, field of view (FOV) = 230 mm × 230 mm, voxel-size = 1.8 mm × 1.8 mm × 3.0 mm, diffusion-weighted b-value = 1000 s/mm^2^, bandwidth = 1396 Hz/Px. All participants were scanned, and none were excluded after being visually inspected for motion artifacts. Moreover, all imaging data were qualified for further analysis without any brain abnormalities (lesions and dissection).

### 2.3. Behavioral Test

All cognitive tasks were administered prior to MRI scanning. Ten participants, each from the control group and the TCC group, completed cognitive tasks.

The Attention Network Test (ANT), also called the Flanker type test, is based on the attention network theory, which examines the effects of cues and targets within a single reaction time task and measures three independent attention networks: alerting, orienting, and executive control [29]. In this study, we focused on the executive control network, which manages the ability to control our own behavior to achieve intended goals and resolve conflict among alternative responses. In the flanker task (Figure 1), participants are required to identify the direction of a central arrow target flanked by neutral, congruent, or incongruent stimuli (arrows in the same or in the opposite direction as the target, respectively), and the executive attention is calculated as the reaction time of the incongruent condition minus the reaction time of the congruent condition. The behavior scores obtained from this ANT are the mean reaction time and the accuracy, i.e., the percentage of correct responses.

### 2.4. Image Processing

The DTI data were processed by PANDA v.1.3.1 toolbox, running with FMRIB Software Library (FSL) version 6.0.3 and MATLAB 2018a. PANDA (Pipeline for Analyzing braiN Diffusion imAges) is a MATLAB toolbox for pipeline processing of diffusion MRI images, which integrates FSL, Diffusion Toolkit, MRIcron, and POS. PANDA can automatically process brain diffusion images through the following steps: (1) image format conversion: the DICOM files of all participants were converted into NIfTI images using the dcm2nii tool in MRIcron. (2) brain extraction and mask estimation: non-brain area was eliminated, and the brain mask was estimated. (3) crop and eddy current/motion correction: the redundant parts of the images were cropped for memory cost reduction, and the images were corrected for the distortions caused by eddy current and head movements. (4) DTI metric calculation: diffusion tensor fitting and diffusion metrics, including FA, MD, AD, and RD, were calculated using the FSL dtifit. (5) normalization: the diffusion metrics were registered to a standardized template in a standard space through FSL FNIRT. Then, the International Consortium of Brain Mapping template (ICBM-DTI-81) was used to parcel the entire WM into 50 core regions, then the regional diffusion metrics were calculated by averaging the values within each region of the WM atlas.

### 2.5. Statistical Analysis

In this study, statistical analysis was conducted using SPSS Statistics 26.0 (IBM Corp., Armonk, NY, USA). Specifically, χ^2^ tests were conducted for the sex proportion, and a two-sample t-test was employed to examine other demographic characteristics. The difference in diffusion metrics between the TCC group and the control group was checked with a one-way ANOVA. Considering the confounders, i.e., age and education, we further performed an ANCOVA analysis while taking these factors as covariates. Additionally, Pearson correlation was utilized to explore the relationship between diffusion metrics, behavioral test results, and TCC experience. *P* < 0.05 was considered statistically significant, and 0.05 < *p* < 0.08 was considered marginally significant.

## 3. Results

### 3.1. Demographic Characteristics and Behavior Performance

The demographic characteristics of each group are summarized in Table 1. Two-sample *t*-tests and the χ^2^ test showed that there were no significant differences in gender (χ^2^ = 0.673, *p* = 0.412), age (t = 1.149, *p* = 0.258), and years of education (t = −0.435, *p* = 0.666) between TCC practitioners and the healthy controls.

For the RT of ANT, no significant difference was observed between the TCC group and the control group, although the TCC group exhibited a trend toward a shorter mean RT compared to the control group because of the limited sample size (t = 1.227, *p* = 0.236, Figure 2A). In terms of the accuracy of ANT, there was also no significant difference between the TCC group and the control group (control group: 99.0% ± 0.008; TCC group: 99.5% ± 0.006; t = −1.421, *p* = 0.173). However, we conducted the correlation analysis based on the hypothesis that TCC practice might induce a change in ANT performance, which showed that the performance of ANT was negatively correlated with TCC experience (r = −0.659, *p* = 0.038, Figure 2B).

### 3.2. Diffusion Metric Variations

As shown in Table 2, extensive differences were observed in these metrics between the TCC group and the control group. With regard to the FA metric, it was found that there were statistically significant differences in three distinct regions (Figure 3A): the right cerebral peduncle (CP) (F = 4.578, *p* = 0.039), right fornix (FX)/stria terminalis (ST) (F = 4.606, *p* = 0.038), and left FX/ST (F = 4.909, *p* = 0.033). However, after taking age and education as confounders into account, the significance of FA alternations became marginal.

The study also identified that the TCC group significantly differed from the control group in MD in the right corticospinal tract (CST) (F = 6.625, *p* = 0.014), right inferior cerebellar peduncle (ICP) (F = 6.045, *p* = 0.019), and left FX/ST (F = 5.070, *p* = 0.030) (Figure 3B). Specifically, the TCC group demonstrated lower MD values in the left FX/ST region compared to the control group. In contrast, the TCC group exhibited higher MD values in the right CST and the right ICP relative to the control group.

Statistically significant differences in AD were observed between the two groups in several brain regions, including the left CP (F = 5.801, *p* = 0.021), the right CP (F = 6.543, *p* = 0.015), the right CST (F = 7.009, *p* = 0.012), and the right ICP (F = 5.127, *p* = 0.029). As shown in Table 2, the TCC group had significantly higher AD values in all the mentioned regions compared to the control group.

Concerning RD, we also found that the TCC group exhibited significantly higher values in the right CST compared to the control group (F = 6.314, *p* = 0.016). Conversely, significantly lower RD values were detected in the left FX/ST in TCC practitioners (F = 6.026, *p* = 0.019).

### 3.3. Correlation Analysis

The correlation analysis indicated that the FA values in the right CP were positively correlated with the accuracy of ANT (r = 0.701, *p* = 0.024) in the TCC group. In the left FX/ST of the TCC group, MD was negatively correlated with the reaction time (RT) of ANT scores (r = −0.720, *p* = 0.019) and positively correlated with the hours of TCC practice (r = 0.502, *p* = 0.017). Regarding the possible influence of age, we performed a partial correlation analysis while controlling for the age factor. The results showed that FA was still positively correlated with the accuracy of ANT (r = 0.701, *p* = 0.035). As to MD, the positive correlation remained significant (r = 0.492, *p* = 0.024), while the negative correlation became marginally significant (r = −0.654, *p* = 0.056). No significant correlation was detected in the control group (*p* > 0.05).

## 4. Discussion

The current study systematically investigated whether TCC practitioners exhibited unique WM microstructure underlying improved cognitive function. Our findings revealed substantial differences in diffusion metrics between the TCC group and the control group. Specifically, the TCC group demonstratedsignificantly different diffusion metrics in CST, FX/ST, and CP. Furthermore, we found a positive correlation between the FA values in the left CP and ANT performance. These results not only extended our previous work on the association between TCC practice and cortical reorganization of WM but also suggested that TCC practice may contribute to the featured WM microstructural organization that underlies complex cognitive processing.

In this study, we found that extensive differences existed in WM microstructures between TCC practitioners and healthy controls. This result is consistent with previous work. For instance, a whole-brain voxel-based analysis revealed that TCC practice had an effect on the prevention of WM degeneration; specifically, TCC practitioners had higher FA in the splenium of the corpus callosum [28]. The possible explanation might lie in the core component of TCC, aerobic exercise. It is well established that physical fitness or activity (PFA) exerts beneficial effects on WM structure. A systematic review [31] has examined the effects of PFA on the WM of the aging brain and reported that higher levels of PFA were associated with increased FA. The association between general PFA levels and WM integrity may be mediated by various neurobiological mechanisms. For instance, evidence suggests that higher levels of PFA may lead to the proliferation of oligodendrocyte progenitor cells, which intermingle with neurons in the central nervous system and form myelin sheaths [32]. Voss et al. [33] demonstrated that greater aerobic fitness was associated with a greater change in WM integrity in the frontal and temporal lobes, which were particularly susceptible to the damaging influence of aging. A recent study [34] revealed an age-dependent association between PFA and MD in widespread WM regions, suggesting PFA may be a promising strategy to foster resilience against age-related cognitive decline. Therefore, the extensive alterations in WM might be induced by long-term TCC practice, which is characterized by typical physical exercise with low or moderate intensity.

As a multimodal mind–body exercise, TCC shares mental components with meditation and is enacted within a state of awareness and concentration. Increasing evidence demonstrates that meditation can induce some changes in WM microstructure. Previous studies revealed significant FA changes in the corticospinal tract, the uncinate fasciculus, and the temporal component of the superior longitudinal fascicle [35], as well as in the corpus callosum in long-term meditators [36]. Tang et al. [37,38] also detected changes in FA, AD, and RD in WM pathways surrounding the anterior cingulate cortex after integrative body–mind training, including the corpus callosum and anterior and superior corona radiata. Recently, they demonstrated that increased frontal midline theta activity was associated with brain WM plasticity, suggesting that this could be the mechanism for changes in WM following mindfulness meditation [39]. Some studies have also investigated the effects of yoga on WM. For instance, higher FA values were found in yoga practitioners in the WM adjacent to the left posterior insula [40].

In the right CST of TCC practitioners, we found significantly higher AD and RD values when compared to the controls. Regarding AD and RD metrics reflecting axonal integrity and myelin sheath, these results may suggest better axonal integrity and fiber demyelination induced by TCC practice in this region, which was consistent with the previous study [35]. CST is an important motor pathway in the human brain, being responsible for transmitting motor information from the motor cortex to the spinal cord. Its role is particularly important for the execution of discrete voluntary movements, especially those involving the hands and feet [41]. The integrity of the CST is critical for the performance of fine, skillful movements [42,43]. Following a stroke, CST is the most extensively studied tract in the context of motor function [44]. Several studies have demonstrated a strong correlation between the structural integrity of CST and motor impairment. For instance, Lindenberg et al. [45] reported that the differential integrity of all corticospinal motor tracts could serve as a reliable structural marker of motor impairment. DTI-derived estimates of structural damage to the CST have been utilized to predict motor recovery following a stroke [46]. In clinical practice, FA values were used to evaluate the integrity of CST to predict motor outcomes and guide clinicians in selecting the most appropriate therapy [47]. Moreover, CST damage has also been associated with proprioceptive deficits [48]. These findings suggest that this difference in WM microstructure might be associated with the improvement of motor function. Furthermore, we also observed increased MD values in TCC practitioners, which is consistent with both higher and lower MD values reported in age- or disease-related neurodegeneration [49]. Due to the increased volume of the extracellular space or decreased barriers to diffusion in WM [50], increased MD typically is a consequence of neuroinflammation, edema, or necrosis [27]. Interestingly, decreased MD has also been observed in patients with obsessive-compulsive disorder [51].

We also identified three WM regions with marginally significant FA values, including bilateral FX/ST. As the major output tract of the hippocampus, FX, which arcs around the thalamus and connects the medial temporal lobes to the hypothalamus, is a critical component of the classic Papez circuit and part of the limbic system [52]. This discrete WM bundle is also unusual because it can be classified as both a projection tract and a commissure. It has been reported that FX undergoes age-related decline [53] and is sensitive to aerobic fitness. Higher cardiorespiratory fitness has been found to be associated with greater FA in a diverse network of WM tracts, including FX, in older adulthood [54]. FX is known to play a crucial role in various cognitive functions such as learning, working memory, and visuospatial abilities [55]. In fact, a study on patients with bipolar disorder and schizophrenia has shown that significant abnormalities existed in the major WM microstructure of bilateral FX [56]. Moreover, decreased FA or increased MD in FX has been demonstrated as a major neuropathological mechanism in Alzheimer’s disease [57,58,59,60]. In the context of the FX/ST region, our study revealed not only higher FA but also significantly lower MD and RD values in this brain region among TCC practitioners. The conjunction of decreased FA and increased MD has been linked with the accumulation of water and gross tissue loss [61], while decreased FA together with increased RD has been associated with loss of myelin in both human studies [62] and animal models [63,64]. Furthermore, this conjunction of metrics has been reported in acute demyelination in multiple sclerosis [65], as well as in patients with euthymic bipolar I disorder [66]. Thus, it is plausible that TCC practitioners may have less gross tissue loss and less myeline loss in the left FX/ST, indicating a beneficial effect of TCC on WM microstructure.

In the right CP, our study found significantly higher AD values in conjunction with higher FA values with marginal significance. This result may further be interpreted as better axonal integrity, which is consistent with previous research. A recent study investigated the effects of TCC on WM microstructure in healthy adults aged 55 to 65 years and found that the TCC group had higher FA in the left CP [67]. Similarly, a study on children with autism who received mini-basketball skill intervention, a combined cognitive and physical training program, also demonstrated higher FA in the right CP [68]. From an anatomical perspective, CP contains major axon tracts that originate in the cerebral cortex and terminate in the brainstem and spinal cord, mainly composed of inferiorly projecting CST and corticopontine fibers [69]. Diffusion metrices in CP have emerged as a promising biomarker in several neurological diseases. For instance, patients with schizophrenia exhibited significantly decreased FA in CP [70], consistent with the findings reported in studies of first-episode schizophrenic patients by Hao et al. [71] and Cheung et al. [72]. Furthermore, CP is also involved in the sense of proprioception and pain responses. For example, a study investigating patients with neuropathic pain after spinal cord injury found that decreases in MD in the CP were associated with both abnormal pain modulation and motor impairment [73]. Significantly lower FA and higher RD in CP were reported in patients with trigeminal neuralgia [74]. Lieberman et al. examined patients with chronic musculoskeletal pain and detected an association between higher AD and RD in CP [75]. They speculated that CP might contain descending pain-inhibiting fibers, the properties of which could be affected by chronic pain. Intriguingly, our correlation analysis in this region revealed a positive correlation between ANT accuracy and FA values, suggesting that this stronger local connectivity may have beneficial effects on behavioral performance. Other studies on TCC have also shown positive influences on brain function and cognitive performance. For instance, Ji et al. [76] compared the benefits of TCC on executive and non-executive cognitive functions with brisk walking. A recent study [17] employed graph theory to examine the effect of TCC from a network perspective characterized by small-world attributes. The results revealed that long-term TCC training produced significant changes in WM small-world attributes, indicating improvements in the efficiency and transmission of neural data between brain networks. Collectively, these findings suggest that TCC practice could be an effective means of optimizing brain structure and improving cognitive performance in the elderly.

The results should be evaluated within the context of their limitations. Firstly, while the inclusion criterion was carefully controlled in participant recruitment, our cross-sectional study design cannot entirely eliminate the influence of confounding factors such as personal lifestyles or preexisting characteristics of brain structure and function. To address these issues, we recommend conducting a longitudinal study in future work. Secondly, it is important to note that the sample sizes of our participant groups are relatively small. Thirdly, due to time constraints, the behavioral data were only collected from half of our sample. Fourthly, there were confounders in this study, e.g., age and education, which may change the significant levels. These results should be interpreted with caution as they may be sample artifacts. Therefore, further investigation involving more samples is warranted.

Increasing evidence indicates that abnormalities in some WM regions are closely associated with some neurological disorders. WM usually undergoes a gradual transition from healthy to abnormal status, especially during aging. We propose that TCC could be an effective means to prevent cognitive decline or aging-related neurodegeneration.

## 5. Conclusions

This study demonstrates that elderly individuals with TCC practice exhibited extensive differences in diffusion metrics compared to the controls, including CST, FX/ST, and CP. Furthermore, the increased FA values in the right CP were also positively correlated with ANT performance in TCC practitioners. These findings provide neuroimaging evidence that TCC practice may contribute to optimized regional WM microstructure. Although a limited sample size is involved in this cross-sectional study, the results still showed a significant association between regional WM structures and cognitive improvement, implying the potential of mind–body practice in neurodegenerative diseases relevant to cognitive impairments in processing complex information. In the future, a rigorous study with a large sample size is warranted when examining the clinical value of mind–body integrative practice.

## Figures and Tables

**Figure 1 brainsci-13-00691-f001:**
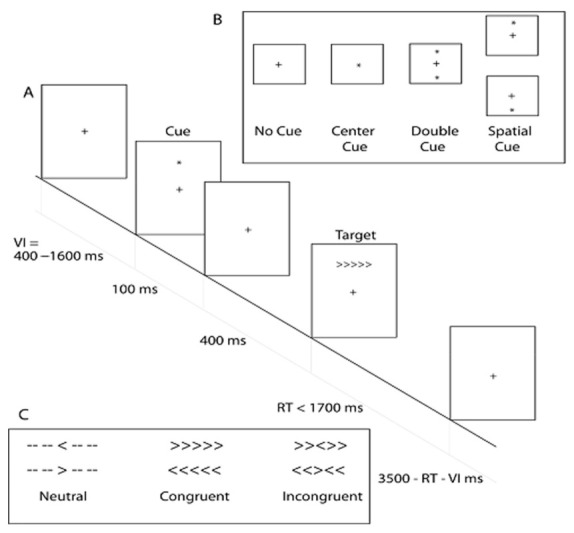
An illustration on ANT. (**A**) A single trial including three stages (**B**) 4 cue conditions: No Cue, Central Cue, Double Cue, and Spatial Cue (**C**) 3 target conditions: Neutral, Congruent, and Incongruent. Reprinted with permission from Ref. [9].

**Figure 2 brainsci-13-00691-f002:**
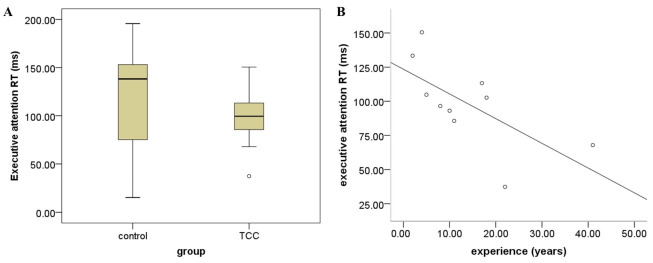
Results of behavior performance. (**A**) RT of ANT comparison (**B**) correlation between TCC experience (years) and RT of ANT. Reprinted with permission from Ref. [30].

**Figure 3 brainsci-13-00691-f003:**
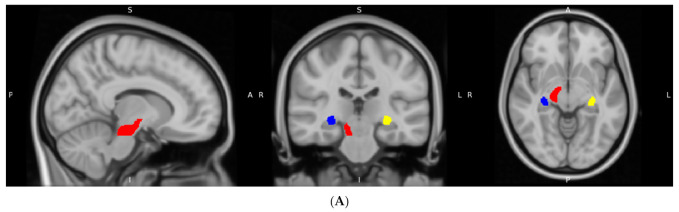
WM regions with significant differences in FA and MD values. (**A**) regions with significant differences in FA (red: right CP; yellow: left FX/ST; blue: right FX/ST) (**B**) regions with significant differences in MD (red: left FX/ST; yellow: right CST; blue: right ICP).

**Table 1 brainsci-13-00691-t001:** Demographic characteristics.

	TCC Group (n = 22)	Control Group (n = 18)
Male/Female	7/15	8/10
Age (years ± SD)	52.4 ± 6.8	54.8 ± 6.8
Education (years ± SD)	12.2 ± 2.9	11.8 ± 2.9
TCC experience (years ± SD)	14.6 ± 8.6	NA

**Table 2 brainsci-13-00691-t002:** Differences in diffusion metrics for specific WM regions.

WM Regions	Mean ± SD		One-Way ANOVA ANCOVA			
	Control Group	TCC Group	F	*p*	F	*p*
**FA**						
right CP	0.6013 ± 0.0199	0.6137 ± 0.0165	4.578	0.039	3.442	0.072
right FX/ST	0.4607 ± 0.0188	0.4776 ± 0.0285	4.606	0.038	3.856	0.057
left FX/ST	0.4690 ± 0.0198	0.4863 ± 0.0280	4.909	0.033	3.665	0.064
**MD (×10^−3^)**						
right CST	0.822 ± 0.056	0.877 ± 0.075	6.625	0.014	5.698	0.022
right ICP	0.758 ± 0.032	0.790 ± 0.046	6.045	0.019	6.025	0.019
left FX/ST	0.852 ± 0.042	0.828 ± 0.028	5.070	0.030	4.188	0.048
**AD (×10^−3^)**						
left CP	1.4886 ± 0.0581	1.5301 ± 0.0573	5.801	0.021	4.688	0.037
right CP	1.4889 ± 0.0543	1.5400 ± 0.0646	6.543	0.015	6.438	0.016
right CST	1.2855 ± 0.0593	1.3427 ± 0.0852	7.009	0.012	6.370	0.016
right ICP	1.1649 ± 0.0420	1.2088 ± 0.0619	5.127	0.029	4.685	0.037
**RD (×10^−3^)**						
right CST	0.5899 ± 0.0593	0.6442 ± 0.0742	6.314	0.016	5.393	0.026
left FX/ST	0.6316 ± 0.0414	0.6024 ± 0.0338	6.026	0.019	4.875	0.034

## Data Availability

Data available on request due to restrictions eg privacy or ethical, The data presented in this study are available on request from the corresponding author. The data are not publicly available due to privacy.

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
