# Peer review of "Enhanced Integrity of White Matter Microstructure in Mind–Body Practitioners: A Whole-Brain Diffusion Tensor Imaging Study"

_brainsci, 2023, doi:10.3390/brainsci13040691_

Round 1

Reviewer 1 Report

The study “Enhanced integrity of white matter microstructure in mind- body practitioners: a whole-brain diffusion tensor imaging study” evaluated the effects of Tai Chi Chuan practice on brain white matter microstructures. Results demonstrated significant differences in DTI parameters between TCC group and control group. Some DTI parameters were also correlated with ANT performances and TCC experience in TCC group.

1.     Is there a between group difference of ANT performances? Please provide results from ANT.

2.     Section 3.1, the gender difference was assessed with two-sample t-test, which was not the correct test for categorical variables.

3.     Based on the description in the method, the atlas contains 50 white matter regions. Are DTI parameters from all of the 50 regions subjected to t-test? If that was the case, for this large number of tests, false discovery rate should be controlled.

4.     Figure 2 is not needed as the results were not from voxel-based analysis.

5.     Section 3.3, MD of left FX/ST was “positively correlated with TCC experience (hours of practice)”. Was it hours of practice or years of practice? If it was hours of practice, please include this information in Section 2.1 and 3.1. In addition, was this correlation driven by age of subjects?

6.     The discussion focused on the aerobic exercise component of TCC and compared the results of this study with other studies on physical exercise. How about the meditative component of TCC?

Reviewer 2 Report

Title: Enhanced integrity of white matter microstructure in mind-body practitioners: a whole-brain diffusion tensor imaging study

The paper presents an empirical study investigating potential effects of Tai Chi Chuan on brain structure.

The research topic is of relevance for both research and practice.

The manuscript is well written and the results of interest.

Important strengths include the detailed assessments of brain structures.

The paper could nicely fit into the Special Issue “Physical Exercise-Driven Brain Plasticity”.

However, the sample contains only 22 individuals that are split into 2 groups.

Are estimates reliable with such a small sample?

Has an a priori power analysis been conducted?

In relation to this major limitation, the findings are verbally overselled at several places throughout the manuscript.

The conclusions need to be softened.

Why is the sampling not balanced?

Which randomization procedure has been used?

Commonly, it is better to have an identical number of participants in each experimental group.

Has the study been pre-registered?

Please include this information.

Several important confounders such as sex, age, health status, education, etc. could not be tested in detail.

The conceptual implications could be elaborated in more depth.

For instance, it is not clear how this study with healthy individuals can tell something on pathological development.

The practical relevance needs to be illustrated in more detailed examples from everyday life.

Round 2

Reviewer 1 Report

The revision addressed most of my comments. Figure 3 looks like screenshots from the display of FSLeyes. It is recommended to make the figure more visually appealing by removing unnecessary blank space, unifying the size of images, and removing crosshairs.
